# The Economic Impact of Parasitism from Nematodes, Trematodes and Ticks on Beef Cattle Production

**DOI:** 10.3390/ani13101599

**Published:** 2023-05-10

**Authors:** Tom Strydom, Robert P. Lavan, Siddhartha Torres, Kathleen Heaney

**Affiliations:** 1MSD Animal Health, 20 Spartan Road, Isando, Kempton Park 1619, South Africa; tom.strydom@merck.com; 2Merck & Co., Inc., 126 E. Lincoln Avenue, Rahway, NJ 07065, USA; 3Merck Animal Health, 2 Giralda Farms, Madison, NJ 07940, USA; siddartha.torres@merck.com (S.T.); monaheaney@yahoo.com (K.H.); 4Heaney Veterinary Consulting, 303 Fletcher Lake Avenue, Bradley Beach, NJ 07720, USA

**Keywords:** beef, parasites, nematodes, trematodes, ticks, prevention, treatment, parasiticide, acaricide, endectocide, economics of treatment

## Abstract

**Simple Summary:**

Cattle parasites live inside or on the body of beef cattle. The most common beef parasites include intestinal roundworms, flatworms and ticks. The act of parasitizing cattle reduces the health of the animals and reduces the economic value to the farmer through reduced body weight, milk production, coat and hide quality and ability to give birth to healthy calves. As a result, beef cattle producers lose billions of dollars in the value of their herds each year due to parasitism. Preventing and treating parasites is an important step in increasing the farmers’ ability to raise healthy beef cattle, make a profit and meet the world’s need for sustainable protein and other cattle products.

**Abstract:**

Global human population growth requires the consumption of more meat such as beef to meet human needs for protein intake. Cattle parasites are a constant and serious threat to the development of the beef cattle industry. Studies have shown that parasites not only reduce the performance of beef cattle, but also negatively affect the profitability of beef agriculture and have many other impacts, including contributing to the production of greenhouse gases. In addition, some zoonotic parasitic diseases may also threaten human health. Therefore, ongoing cattle parasite research is crucial for continual parasite control and the development of the beef cattle industry. Parasitism challenges profitable beef production by reducing feed efficiency, immune function, reproductive efficiency, liveweight, milk yield, calf yield and carcass weight, and leads to liver condemnations and disease transmission. Globally, beef cattle producers incur billions (US$) in losses due to parasitism annually, with gastrointestinal nematodes (GIN) and cattle ticks causing the greatest economic impact. The enormity of losses justifies parasitic control measures to protect profits and improve animal welfare. Geographical differences in production environment, management practices, climate, cattle age and genotype, parasite epidemiology and susceptibility to chemotherapies necessitate control methods customized for each farm. Appropriate use of anthelmintics, endectocides and acaricides have widely been shown to result in net positive return on investment. Implementing strategic parasite control measures, with thorough knowledge of parasite risk, prevalence, parasiticide resistance profiles and prices can result in positive economic returns for beef cattle farmers in all sectors.

## 1. Introduction

The global human population continues to increase at a rate of slightly more than 1% per year [1]. With natural resources remaining unchanged, feeding this increasing population in a sustainable way has become a pressing issue in many parts of the world. Meat consumption is an increasing source of nutrition for many people worldwide. As global population and income increase over the next decade, consumption of meat is expected to increase by 14% by 2030 compared to average consumption from 2018–2020. Specifically, beef production is expected to grow 5.9% to supply a portion of this increasing demand for meat [2].

There are many challenges to profitable beef production, not the least of which is parasitism. Parasites and parasite-borne diseases negatively affect cattle and also influence the marketing and trade (national or international) of animals and food products [3]. Parasitic infections reduce liveweight, milk yield and feed efficiency, and are a leading cause of liver condemnations, mortality in young cattle and, in some parts of the world, even mortality in adult animals [4,5,6,7,8,9,10,11]. Parasitic infestation can also negatively impact reproduction and immune response to vaccination and disease [12,13,14,15,16,17,18].

To maintain profitability, measures implemented to mitigate the impact of parasitism must result in a positive economic return on investment for cattle producers. Methods for successful parasite control change over time; parasite susceptibility to chemotherapeutics shifts, new management strategies are introduced, and parasite epidemiology fluctuates [3]. Therefore, periodic review of the literature on parasite control is useful for directing contemporary mitigating strategies. To assist beef producers and veterinarians, this paper aims to summarily review current trends on the impact from the most economically significant parasite species across beef cattle production in temperate and subtropical areas, specifically considering return on investment. Table 1 lists common cattle parasites of economic importance.

## 2. Beef Cattle Parasitism: Estimating Losses in a Changing Landscape

The world’s largest beef and veal producers are, in order, the United States of America (USA), Brazil, China, the European Union, India, Argentina, Mexico and Australia, which collectively contributed 74% of the estimated total 58,184,000 metric tons of beef carcass weight produced in 2022 [19]. With commodities as large as this, even minor changes in production efficiency can lead to extraordinary differences in profit and loss. For example, in Brazil, the second largest beef producer, the estimated annual beef production losses (in US$ billions) due to gastrointestinal nematodes (GIN), Liver fluke, the cattle tick, *Rhipicephalus microplus*, flies and grubs were US$7.11 Billion (B), US$0.210 B, US$3.24 B, US$2.9 B, and US$0.72 B, respectively, for a combined total lost revenue of US$13.96 B [20,21]. In Mexico, the seventh largest beef producing country, the yearly economic loss due to the six major parasites of cattle was estimated at US$1.41 B [9]. In the European Union, the annual estimated costs of helminth infections in beef cattle were € 0.423 B [22]. In 2015, production losses in cattle due to internal parasites and cattle ticks (*Rhipicephalus microplus)* across Australia were estimated at AUS$0.2546 B per annum [23]. In the USA, the increased cost of beef cattle production due to GIN has been estimated at $190 per head per year [24]. With 44.9 million head of cattle marketed in 2020, losses due to GIN can be estimated at approximately US$8.5 B annually [19]. Liver fluke infestation noted in US cattle at slaughter was 1.1–5.5% of cattle processed. In 2016, offal condemnation cost producers US$2.56 per head [25,26,27]. Losses due to parasitism in cattle are summarized in Table 2.

In addition to financial losses, parasitism impacts the health and welfare of farmed cattle. The World Organization for Animal Health defines good animal welfare as when an animal is “… healthy, comfortable, well nourished, safe, is not suffering from unpleasant states such as pain, fear and distress, and is able to express behaviors that are important for its physical and mental state. Good animal welfare requires disease prevention and appropriate veterinary care, shelter, management and nutrition…” [28]. Low-level gastrointestinal nematode infection has been shown to negatively influence cattle feeding, rumination, resting and standing behaviors and thereby negatively impacts their overall welfare [29,30,31,32]. Tick infestation also impacts cattle welfare, not only in transmitting disease-causing organisms, but as the tick bites it injures tissues at their feeding site, causing irritation, inflammation, hypersensitivity and dermatitis [33], and can confer paralysis toxin [34].

Parasitism may also play a role in global meat production sustainability. Cattle and sheep infected with gastrointestinal nematode and liver fluke appear to contribute greater greenhouse gas emissions than uninfected animals [35,36,37,38,39]. With some governments issuing emission reduction targets, parasite control may become necessary to help satisfy increasing political and societal pressure for cattle producers to reduce their carbon footprint [40]. 

The enormity of losses from parasitism justifies implementation of methods to control the major parasites affecting beef cattle, to protect profits, improve animal welfare and perhaps play a role in reducing greenhouse gas emission. However, cattle operations face narrow operating margins [40,41,42], so any methods employed to mitigate losses from parasitism must be cost-effective. The objective of this treatise is to examine the changing landscape of parasite control and review trends in managing parasitism that optimize production and return on investments, considering input costs for drugs, labor, pasture, and feed, and the economic impact of failure to treat. The scope of this paper is limited to beef cattle production in temperate and subtropical areas. Appropriate use of antiparasitics to treat clinically apparent parasitism to reduce morbidity and mortality has been well described by others [43,44,45,46] and is not the focus of this treatise.

Successful, cost effective and profitable management of parasitism is dependent upon an understanding of the many variables that influence the level of infection. The types of parasites, the extent of their effects on cattle, and measures to control them differ by geography, climate, cattle genotype, age of cattle, production environment, and management practices. For example, tropical and subtropical environments are usually ideal for the life cycle of several parasites. As a result, the prevalence and variety of parasitic diseases in those areas are substantially greater than are those in temperate climates [3,47,48,49,50,51]. Consequently, parasite control methods developed for use in temperate areas will prove unsuccessful in other climates [50].

Cattle genotypes differ in their susceptibility to various parasites. For example, *Bos indicus* cattle tend to be more resistant to tick infestation than *Bos taurus* breeds [7,52,53,54,55]. *Bos taurus* breeds have been suggested to be more resistant to GIN than *Bos indicus* breeds [56,57], although others could not substantiate this finding or have shown them to be more susceptible [53,55].

Young cattle, especially those on pasture for the first time, tend to have higher rates of GIN infection than adult cattle [57,58,59]. Even individuals within the same cattle class and management system will have varying degrees of inherent resistance to nematodes and ticks [60,61,62,63]. Cattle on pasture have greater exposure to nematodes and flukes than do cattle in feedlots [46,64,65,66,67,68]. Seasonality and nutritional status also affect cattle exposure and response to parasites [50,60,69].

The distribution of some parasites, and oftentimes the diseases they vector, is expanding because compatible environmental conditions that support their life cycle are expanding. Examples are the broader distribution of the tropical cattle tick (*Rhipicephalus* [*Boophilus*] *microplus*) in West Africa, and the recent (2017) discovery in the United States of the Asian longhorned tick, or bush tick, *Haemaphysalis longicornis* [70,71]. This tick is a vector for *Theileria orientalis*, which causes production losses and death in cattle in other countries and could be a vector for parasite-borne pathogens in the USA [72]. In Brazil, a comparison between 30 years of climatological data and *Rhipicephalus microplus* population dynamics indicated an increase in environmental temperature may be a determining factor in the increased numbers of ticks [69]. In portions of Europe, measurable changes in overall abundance, seasonality, and spatial spread of endemic helminths, attributed in part to climate change, have been predicted and observed [73,74,75,76]. In a study of the financial impact of fascioliasis in 160 Scottish livestock farms with and without climate change, Shrestha et al. (2020) found a 6% reduction in net profit on an average beef farm under standard disease conditions without the effects of climate change. When the effects of climate change were added, losses increased six-fold [77]. Clearly, climate change will have an impact on the profitability of cattle production.

While climate is an important driver of tick and helminth distribution, husbandry practices on the farm also impact parasite transmission to livestock [3,50,66,78,79]. Pasture management administered with knowledge of local parasite epidemiology can successfully limit cattle parasite exposure [3,4,43,46,66,80,81,82]. Strategies such as pasture rotation with crops, annual or biannual pasture renovation and co-grazing with alternate, less-suitable hosts can gradually reduce parasite contamination [3,49,66,83]. Limiting cattle exposure to the snail intermediate hosts by draining wet pastures or restricting access to them has been shown to reduce liver fluke infection [80,81]. Nutritional management can also reduce the severity of parasite infection or enhance immunity [49,84,85,86].

Worldwide, reports of parasite resistance to anthelmintics, endectocides and acaricides are increasing [79,87,88,89,90,91,92,93,94,95]. In Europe, the United States of American, Australia, New Zealand, Brazil and elsewhere, helminth resistance to macrocyclic lactone has been widely demonstrated [90,93,96,97,98,99,100,101,102,103,104]. Anthelmintic resistance to benzimidazoles and imidazothiazoles have also been reported in North America, Australia, Brazil and Europe [87,90,93,105,106]. Anthelmintic resistance has even been identified in farms with no or low treatment history and without any epidemiological or trade links [99,103]. Control of liver fluke has long been challenging [15]. There are few drugs available to treat liver flukes and many of these have low efficacies against damaging juvenile stages of *Fasciola hepatica*. Additionally, now there is evidence of drug resistance in liver flukes [15]. Widespread parasiticide resistance will necessitate the implementation of integrated pest management (IPM), wherein the systematic combination of multiple pest control strategies is employed to achieve sustainable and profitable parasite control [82,107].

In addition to changes in parasite distribution and susceptibility, changes in consumer preferences are impacting beef production [3,24,108]. In some parts of the world, there has been a shift in consumer preferences towards “natural” or organically produced beef [108]. A portion of consumers are willing to pay a premium for these products. Beef production without the benefit of pharmaceutical technologies can be more costly [24]. Nonetheless, a portion of beef producers will want to satisfy this market and institute “natural” means of parasite control, employing non-pharmacological control methods such as pasture and habitat management, stocking rate and breeding for host resistance [61,109,110,111,112,113].

Overall, changes in climate and its impact on parasite distribution, anthelmintic, endectocide and acaricide resistance and consumer preferences will necessitate modifications in the way in which parasite control has been managed in the past [3,63].

## 3. Parasites of Economic Importance

Gastrointestinal and pulmonary nematodes, liver flukes (trematodes), ticks, flies, lice, and mites all cause economic losses and negatively impact animal health and welfare [3,9,10,20,22,77,114]. In addition, many of these pests serve as vectors for viral, protozoal and bacterial organisms that can cause devasting diseases such as anaplasmosis, babesiosis, East Coast fever caused by *Theileria parva* and heartwater caused by *Ehrlichia ruminantium* [44,115,116]. As shown above, cattle parasites causing the greatest economic losses in beef cattle production worldwide are nematodes, trematodes and ticks (Table 1).

### 3.1. Nematodes—Major Species, Health Impact and Economics of Control

Nematode parasites are one of the most common and important limiting factors that affect the health and wellbeing of livestock [43]. Gastrointestinal and respiratory nematodes (GIN) of cattle live on pasture and in their host. Adult worms live and reproduce within the animal. Eggs produced by these worms pass in the fecal material and contaminate the pasture. The eggs then hatch and develop into the infective stage on the pasture where cattle become infected from ingesting infective larvae when grazing [43]. Once ingested, worms establish themselves in the gut lumen where they actively feed either on blood, as is the case with *Haemonchus placei*, or on other tissues and fluids from the mucosal surface, as is the case with *Ostertagia ostertagi*. The infection results in decreased appetite, decreased gut retention time, and a net fluid, electrolyte, and nutrient loss to the gut lumen, which negatively impacts feed intake, growth rate, carcass weight, carcass composition, fertility, immune response and milk yield [43,117]. When infection is severe, inflammation in the gastrointestinal tract can lead to parasitic gastroenteritis (PGE). Cattle suffering from PGE have watery diarrhea, anorexia, poor haircoat, and loss of body condition. More often, infection with GIN does not necessarily cause overt clinical disease, but rather subtle changes in production efficiency [65]. These production changes can be difficult to recognize but result in economic losses to the producer. For example, studies of stocker cattle showed significant increases in average daily gain (ADG) for calves treated with anthelmintics compared to controls [118,119]. Similar advantages in calf weaning weight have been reported for anthelmintic treated cows and heifers compared to controls [13,14]. Anthelmintic treatment has also been associated with improved reproductive performance with significant increases in pregnancy rate in anthelmintic treated cows and heifers compared to control animals [13,17,18,120]. Using the differences in average ADG for parasite free calves, the value of calf-weight sold, the number of days on pasture, the cost of anthelmintic and handling costs can be used to calculate individualized increased profit resulting when parasites are removed [43,104]. Studies comparing high and low efficacy anthelmintics indicate financial gains with high efficacy treatments [85,86] whereas it would be more effective not to treat than to use low-efficacy drugs [86]. Better cost effectiveness resulted from improved animal production indices when proper nutrition was combined with an anthelmintic treatment that effectively reduced nematode egg shedding [85].

Infection with the respiratory nematode, *Dictyocaulus viviparus*, or lungworm, can also negatively impact the health, welfare, and production efficiency of beef cattle. This parasite is found sporadically in pasture-raised cattle in Europe, North and South America, Brazil and Australia [117,121,122,123]. Data on prevalence in beef cattle are limited. However, data in dairy cattle from across Western Europe indicate variable infection rates range from 3% in Switzerland [124] to 17% in Germany [125], 20% in Belgium [126], 63% in Ireland [127] and 80% in the Netherlands [128]. Sero-epidemiological surveys in first grazing season (FGS) calves in Sweden (dairy and beef) showed farm prevalence of approximately 40% [121,129,130].

Cattle become infected with lungworm by ingesting third-stage larvae from pasture [121]. These larvae migrate in the vasculature to the lungs, leave the branches of the pulmonary artery and migrate through the lung parenchyma to the airways. This migration and the host inflammatory response results in difficulty breathing, coughing, noisy lung sounds, emphysema and rapid loss of condition. Infection with *D. viviparus* can also reactivate previous infections such as bovine rhinotracheitis (Bovine Herpes Virus 1) infections which can complicate the clinical signs, often leading to profuse nasal discharge [131]. The respiratory distress caused by lungworm is a serious animal welfare problem. Clinical lungworm disease is most often seen in young animals towards the end of their first grazing season. Mild infections result in subclinical disease [122,132]. While reports of economic losses attributed to lungworm infection in beef cattle are scant, research in pastured dairy herds indicate significant economic losses associated with reduced milk yield, repeated insemination and higher treatment costs [133,134].

In Europe, 81% of the annual economic losses due to helminth infections, including GINs, lungworms and liver flukes, was due to lost production and 19% was due to cost of treatment [22]. The use of broad-spectrum anthelmintics has long been the go-to for the treatment of nematode infection in cattle [3]. With the introduction of truly broad spectrum anthelmintics, including the benzimidazoles, the pro-benzimidazoles, the imidazothiazoles and the tetra-hydro-pyrimidines in the 1960s, farmers achieved considerable success in the control of cattle nematodes [3]. The introduction of the macrocyclic lactones (ML) in the 1980s was a momentous improvement in nematode and ectoparasite control because the potency, convenience, spectrum and eventual low cost of the MLs largely replaced the need for critical thinking about parasite control [89,113]. Routine treatment with MLs became the default parasite control “strategy”. In a survey of cow-calf producers in Western Canada, Wills et al. (2020) found routine dependency on pour-on ML control of external parasites [135]. While not administered for control of internal parasites in this case, the approach exerts substantial pressure to select for anthelmintic resistance (AR) [87,89,98,113].

Anthelminthic resistance (AR) or lack of efficacy in nematodes of beef cattle are emerging issues globally with implications for effective parasite control. The lack of efficacy of ML has been reported in bovine farms in Australia, Belgium, Brazil, Denmark, France, Germany, Ireland, New Zealand, the UK, Sweden and Spain [90,93,97,99,101,102,103,105,136]. ML resistance results in significant production losses and treatment costs annually [22]. Across Europe, aggregated farm level prevalence of AR was 8% for benzimidazoles (BZ), 32% for MLs, 12% for levamisole (LEV) and 27% for moxidectin (MOX) [137].

AR means that farmers are no longer reliant on the routine use of single class chemotherapeutics for effective helminth control. Treatment failure due to AR can have greater economic impact on cattle producers than no treatment because treatment costs are added to continued production losses [86]. Alternative strategies for effective and sustainable helminth control, designed to limit production losses due to nematode infection while maintaining the efficacy of available anthelmintics, have been investigated [138,139,140,141,142]. Two such strategies are targeted treatment (TT), wherein the whole herd is treated based on knowledge of the risk or severity of infection, and targeted selective treatment (TST), wherein only certain individual animals within the herd are treated, such as those with high parasite loads or poor parasite tolerance, or simply animals randomly selected for treatment, with the remaining herd left untreated. The untreated animals contribute anthelmintic susceptible parasites (known as refugia) to the pasture. This population of susceptible parasites is intended to help slow AR selection pressure and thereby prolong anthelmintic efficacy [138,139,140,141,142].

Strategies that employ a timely combination drug therapy have also been reported with success [67,93,143,144,145,146]. Walker et al. (2013) found that the administration of a benzimidazole with a macrocyclic lactone given at two different times provided GIN control and improved weight gains for stocker calves grazing warm-season pastures [143]. Fiel et al. (2017) found that a pasture population of ivermectin-resistant *Cooperia* spp. could be replaced by a susceptible one based on similar refugia management. Cattle were monitored monthly for GIN and treated with levamisole when needed. In this study, the clinical efficacy of ivermectin increased from an initial 73% to 99.4%, while the absolute efficacy increased from 54.1% to 87.5% after just two animal production cycles [146]. When resistance to avermectins was found in US pastured stocker cattle, a combination of eprinomectin and levamisole was found most effective against both intestinal and abomasal nematodes, whereas levamisole treatment alone was not effective against *Ostertagia ostertagi* [98]. Similar results were demonstrated by Smith (2014) [144]. In the subtropical climate of the Rio Grande do Sul in Brazil, a study of intensively reared pasture cattle harboring multi-drug resistant *Cooperia* spp., *Trichostrongylus* spp. and *Haemonchus* spp., two-drug combinations were found to be effective against nematode populations identified as resistant to the same compounds when used individually [93]. The most effective combinations were moxidectin plus levamisole, doramectin plus fenbendazole and levamisole plus closantel [93].

Knowledge of the unique AR landscape on individual farms will be necessary for devising economically sound deworming strategies [57,142]. Fecal monitoring can be a practical tool for parasite management on many farms. For others, test methods to manage AR effectively and economically or implement test and treat strategies must be simplified to encourage widespread adoption [107,142,147].

Production and financial gain following nematode treatment have been demonstrated for all sectors of beef cattle production, in temperate, subtropical and tropical climates [7,13,17,22,24,43,67,85,86,99,145,148,149,150,151,152,153,154]. For example, results from a meta-analysis of over 170 research trials utilizing 20,000 MonteCarlo simulations to evaluate the impact of various pharmaceutical technologies on the cost of cattle production revealed that elimination of deworming resulted in increased cattle production cost of US$190 per head over the lifetime of the animal, based on 2005 US production costs and sale prices [24]. Cost saving on production were greatest for cow-calf operations, with savings of US$165 per head. Eliminating de-wormer use in stocker and feedlot cattle resulted in increased production costs of US$21 and US$22 per head, respectively. In a study of 6320 Nelore cows on pasture, the return on investment (ROI) for GIN treatment with moxidectin in association with estrus synchronization, considering all input costs and the economic benefit from increased pregnancy rate and marketed calf value, was 44.9 and 19.0 for primiparous and multiparous cows, respectively [152]. In Brazilian grazed cattle, Conde et al. found that each US$1 spent on deworming, including product and labor costs, resulted in a return of US$157 and US$134 using two or three annual doses of anthelmintic, respectively [155]. Nakatani et al. found that GIN treatment of Brazilian feedlot cattle with fenbendazole resulted in a net return of US$14.60 per animal [154]. The benefit of GIN treatment remained viable when modeled under optimistic, probable and pessimistic financial scenarios relative to cattle pricing [152,153,154,155].

### 3.2. Cattle Trematodes (Flukes)—Major Species, Health Impact and Economics of Control

Liver flukes, also known as trematodes, are flat worms that parasitize the liver of several species of animals including cattle [156]. These parasites are found in more than 70 countries worldwide. Prevalence in cattle is variable, with ranges of 1.2–91% in Africa, 3–67% in the Americas (ex-USA), 1–69% in Asia, 26–81% in Oceania, and 0.12–86% in Europe [114]. The most commonly encountered species is *Fasciola hepatica*, which is predominately found in temperate climates, but can also be found in tropical and subtropical countries, including those in the Middle East, South America and Asia. Another liver fluke species that infects cattle, *Fasciola gigantica*, is found in tropical climates in less developed regions throughout Asia, Africa and the Middle East [156]. Unlike nematode parasites, which have a direct life cycle, *Fasciola* spp. need a lymnaeid snail as an intermediate host. The parasites develop in the snail over many weeks, emerge from the snail and attach to vegetation where they are ingested by foraging cattle. Once in the cow, juvenile flukes migrate through the peritoneal cavity, penetrate the liver and migrate there for up to eight weeks before entering the bile duct. Within the host, immature flukes cause trauma and inflammation as they migrate through the liver, and upon entering the bile duct, they cause obstruction and cholangitis. Liver flukes complete their life cycle in 4.5 to 6 months, at which time they begin to shed eggs and contaminate pastures [64].

The geographical distribution of *Fasciola* spp. is limited to areas where the appropriate snail species is present [64,157]. These are areas with high annual rainfall, large areas of poorly drained pasture, and certain soil types that provide suitable snail habitat. Timing of fluke transmission varies in accordance with seasonal rainfall in each geographic region where both flukes and snails are found [81,158].

Fasciolosis causes huge financial losses to butchers, farmers, and consumers in the form of liver condemnation, poor quality carcass, reduction in growth rate and reduced productivity [114]. *Fasciola* spp. can cause significant morbidity and mortality in beef cattle. However, more often liver fluke infection causes subclinical disease with resultant reductions in feed efficiency, growth, fertility (delayed puberty and increased calving interval) and overall loss of productivity [45,64,68,156,159,160]. Economic losses result from reduced cattle weights across all age ranges, lowered reproductive performance in brood cows and increased culling, reduced milk production leading to lower calf weaning weights and reduced growth weights in stocker cattle [64,68,160] Losses in the feedlot attributed to liver fluke infection result from reduced feed-conversion ratios, lowered average daily gains and substantial delay in reaching slaughter weight [64,160,161]. As few as one to 10 liver flukes have been found to increase slaughter age for animals [160]. In one UK study, cattle with fasciolosis took on average 10 days longer to reach market weight than animals with no evidence of fasciolosis [160]. At the abattoir, liver fluke infection causes increased liver condemnations at slaughter, lighter carcass weights, lower levels of fat and poorer carcass quality scores, resulting in reduced price paid to farmers [159]. In a study of 160 Scottish farms, liver fluke infection caused an average 6% reduction in profitability on an average beef farm [77]. On Swiss farms, liver fluke infection resulted in economic losses from liver condemnation, reduced meat production and decreased fertility [162]. Worldwide annual losses in cattle due to liver fluke infection have been estimated at US$3.0 B [9,94,114,157,160].

Liver fluke control is only relevant for cattle raised in or purchased from areas where both liver fluke and the snail intermediate host reside. Liver fluke control programs must be customized according to local parasite transmission times, those times of year when pasture temperature and moisture are conducive to development of the snail and fluke, and incorporate an effective flukicide, efforts to reduce snail populations and grazing management to limit liver fluke exposure [64]. Pasture management to avoid grazing in wet areas when infected snails are present will prevent infection [81]. Since available flukicides have varying efficacies against juvenile and adult flukes, knowledge of the time of infection is critical to treatment selection [45]. The choice of flukicide and time of use will depend on whether the treatment goal is therapeutic, to improve the health and productivity of the herd, or strategic, to stop liver fluke egg output and reduce pasture contamination [94]. The clinical and subclinical effects of liver fluke infection can be influenced by stocking rate, forage quality, and concomitant infection with GINs, so these factors also need to be considered when designing a fluke control program [14,64].

Triclabendazole is the most widely used anti-fasciola drug [94]. Reports of triclabendazole resistance in *F. hepatica* and *F. gigantica* from cattle occur worldwide and are increasing [94,163]. Resistance has also been reported for other anti-fasciola drugs, including albendazole, clorsulon, closantel, oxyclozanide, and rafoxanide [94]. For this reason, farm management practices, along with knowledge of drug resistance in resident liver fluke populations, are needed so that drugs can be used wisely, and their efficacy conserved. Diagnostics that quantify or estimate fluke burdens could help predict production losses and can be used to guide treatment thresholds [68].

Climate change can influence the free-living stages of *F. hepatica* and its snail intermediate host and can impact the risk of exposure. For example, climate projection data show unprecedented levels of future fasciolosis risk in parts of the UK [74]. Models of financial impact from fasciolosis in beef cattle have predicted a six-fold increase in losses when climate change is included in the analysis [77].

Production gains following treatment for liver fluke have been well demonstrated [14,22,68,164,165,166,167]. In a review of 1582 published studies, Hayward et al. found treating flukes resulted in positive effects on daily weight gain, live weight and carcass weight of 9%, 6% and 0.6%, respectively. The authors also found that younger animals infected with flukes tended to have more severe weight gain and that fluke infection tended to worsen live weight over time [68]. In a large survey of cattle producers in Florida, control of liver fluke resulted in 8 to 10 kg heavier cull cows, 1% to 3% more calves, and 14 to 20 kg heavier calves at weaning, yielding a net return to the producer of US$15.19 to US$31.03 per brood cow, depending on size of the calf crop and calf prices [165]. In a study of a commercial cow–calf operation in Louisiana, calves from cows receiving treatments for both flukes and nematodes had an average weight gain advantage of 8.9 kg in 205-day adjusted weaning weights compared with that of calves from cows receiving treatment for nematodes alone [158]. Similarly, Loyacano et al. demonstrated treating heifers for both nematodes and liver fluke resulted in higher body condition scores and pregnancy rates than heifers treated for nematodes alone [14]. In an area of Southeast Asia where *F. gigantica* prevalence exceeds 30%, control of fasciolosis resulted in a significant average net benefit of US$60 per animal per year [167]. This economic net benefit increased as animals got older. This work demonstrated that a fasciolosis control program, even a relatively expensive one, is economically viable in areas of high risk for fasciolosis.

### 3.3. Cattle Ticks—Major Species, Health Impact and Economics of Control

About 80% of the world’s cattle are affected by ticks and tick-borne diseases (TBD), both of which cause significant production losses [112]. Ticks of economic importance to cattle production exist worldwide, are broadly found in tropical and subtropical areas of the world [34,168,169] and have been summarized in Table 1. Cattle in Asia, Australia and Central and South America are affected primarily with *Rhipicephalus (Boophilus) microplus*, whereas cattle across Africa are affected by species from *Rhipicephalus*, *Amblyomma* and *Hyalomma* [115,170,171]. Ticks cause severe economic losses through the direct effect of tick attachment with the resultant “tick worry” and lost productivity, by injection of toxins and resultant “tick paralysis”, by blood loss from tick feeding, and indirectly by their vectoring of disease-causing pathogens [34,42,61,172]. “Tick worry”, or the unease and irritability experienced by cattle when severely infested with ticks, often leads to serious loss of energy and weight, affecting not only production economics, but animal welfare as well. Production losses per tick and for various cattle breeds have been calculated [173,174,175,176]. For example, in a review of 19 papers, Jonsson et al. (2006) estimated daily production losses attributed to each *R. microplus* engorging female tick as approximately 1.37 g bodyweight in *B. taurus* cattle and 1.18 g bodyweight in *B. taurus* x *B. indicus* cattle. These values were not statistically significantly different, indicating that the treatment threshold number of ticks would be the same for these types of cattle [176]. In a study of *R. microplus* control and its effect on beef cattle performance in the Brazilian Cerrado, Calvano et al. (2019) showed that tick infestation resulted in reduced weight loss equivalent to US$34.61 per animal in the backgrounding phase and US$7.97 per animal in the finishing phase for Brangus animals and crosses [172].

Ticks transmit a diverse array of pathogens including protozoa, bacteria and viruses [11]. Tick-borne pathogens that affect cattle are among the diseases listed as notifiable by the World Organization for Animal Health. These include bovine babesiosis, anaplasmosis, theileriosis and heartwater (*Ehrlichia ruminantium*) [11,78,177].

The cattle fever tick, *R. microplus*, is considered to be the most economically important ectoparasite of livestock worldwide [11,34]. This invasive tick species is the vector of *Babesia bovis* and *Babesia bigemina*, causing babesiosis in cattle in tropical and subtropical parts of the world. Babesiosis causes significant illness in cattle with clinical signs that include anemia, fever, hemoglobinuria and death. Estimates place bovine babesiosis at the top of arthropod-borne diseases causing financial losses for cattle producers [11,34]. A meta-analysis of over 81,000 samples from 62 countries across six continents revealed an overall global prevalence of bovine babesiosis as 29%. Prevalence regionally was 52% in North America, 64% in South America, 61% in Australia, 22% in Europe, 27% in Africa and 19% in Asia [178].

Other tick species also transmit TBD agents of economic importance. The agent responsible for bovine anaplasmosis is vectored by over 20 tick species, mainly *Rhipicephalus* spp. and *Dermacentor* spp., and is common throughout tropical and subtropical regions worldwide [179]. Anaplasmosis causes abortion, weight loss, lost production and adult cattle death [6]. In Tanzania for example, economic losses from *Rhipicephalus appendiculatus*, which is the vector of a protozoan parasite, *Theileria parva*, causing theileriosis, also known as East Coast Fever, has a major impact on livestock farming in sub-Saharan Africa, killing over one million animals each year [6,180]. In one study, each engorging female tick was associated with a loss of 4 g body weight per day in *B. taurus* cattle [181].

There is evidence that ticks and TBD are increasing [178]. Climate change, with the resultant movement of tick populations and transportation of cattle from endemic to non-endemic areas, has facilitated the spread of TBD [179,182]. Native tick species are expanding their regional range and exotic species have been identified in unfamiliar geographies [70,183]. For example, the lone star tick is expanding its range in the United States (US), and the Asian longhorned tick (*Haemaphysalis longicornis*) and the red sheep tick (*Haemaphysalis punctata*) have been identified for the first time in the Western Hemisphere [71]. *R. microplus* was discovered in Ivory Coast in 2007 and then gradually was found in other countries in West Africa, replacing indigenous tick species in the region [70,183]. The introduction and expansion of *R. microplus* puts more cattle at risk for babesiosis.

The importance of ticks and TBD in cattle production is evidenced by the many efforts of different countries to eradicate certain ticks and TBDs [11]. For example, in 1906, the United States government initiated a *R. microplus* eradication program intended to eliminate bovine babesiosis [184]. At that time, production losses from babesiosis were estimated at US$63.25 million, annually. After the ticks were eradicated, the subsequent savings were estimated to exceed US$3 B annually [184]. Return on investment was US$98 for every US$1 spent [8]. In 1996, worldwide economic losses from ticks and TBD were estimated to range from US$13.9 B to US$18.7 B per year [185]. By mid-2000 global estimated losses due to ticks and TBD ranged from US$20 B to US$30 B per annum [186]. Regionally, losses from ticks and TBD were estimated at US$3.24 B and US$0.5736 B per year in Brazil and Mexico, respectively [9,20].

Tick control has relied primarily on the strategic use of acaricides [168]. These chemicals provide rapid and cost-effective tick control. However, routine and indiscriminate use, and inaccurate dosing, have contributed to acaricide resistance worldwide, including cross-resistance and multiple drug resistance (MDR), especially in *R. microplus* [82,95]. Resistance to most chemical classes, including organochlorines, organophosphates, carbamates, formamidines, pyrethroids and ML, has been reported [79,83,95].

The occurrence of acaricide resistance requires more nuanced tick control measures. Alternative methods of tick control, including the use of naturally tick-resistant cattle, biological control (biopesticides), grazing management for tick population control, and tick vaccines, have also been employed with variable success [61,112,172,182,186,187]. Breeding cattle for genetic resistance to ticks and TBD is promising but lacks rapid results [61]. Improved nutrition has been shown to mitigate some of the negative effects of tick infestation [84]. The rotational use of acaricides with differing modes of action and the use of acaricide combinations along with tick population monitoring and pasture management may help preserve the efficacy of existing compounds [188,189]. Studies suggest some small-scale farmers in Africa are unaware of acaricide chemical class differences and consequently fail in their attempts to rotate or combine different chemical classes. This also encourages resistance development [79]. The availability of simple-to-use methods for resistance monitoring, education on acaricide mode of action and accurate dosing and application will be critical to the successful use and maintaining long-term effectiveness of acaricides [79,82,83]. A combination of various tick control methods, individualized for each farm, will provide the best protection against ticks and TBD in cattle [182].

High tick infestation in cattle has been shown to result in lower body weight [190]. Economic thresholds that justify acaricide treatment based on acaricide cost, the price of beef, and weight loss due to each engorging tick have been calculated [173,174,176]. In one such estimate, using the 2005 beef price of US$1.75 per kg liveweight, acaricide cost of US$4.50, including administration, and an average liveweight loss per engorging tick of 1.25 g per day, Jonsson et al. (2006) estimated a mean of at least 100 ticks is necessary for treatment to be profitable [176]. In a study of *R. microplus* infestation in Brangus and Nelore cattle, Calvano et al. found that economic losses of US$34.61 per animal in the backgrounding phase and US$7.97 per animal in the finishing phase occurred for Brangus cattle. The cost of treatment ranged from US$0.55–2.25 per animal depending on the treatment used. Tick control, irrespective of treatment class (organophosphates, pyrethroids, or ML) or method of application (spray, pour-on or injection) was financially rewarding. Return on investment for Nelore cattle was lower than for *B. taurus* breeds due to the lower weight loss associated with tick infestation in this group [172]. Using bioeconomic modeling, Calvano et al. (2019) investigated the profitability of controlling *R. microplus* on cattle compared to no tick control in extensive, semi-intensive and intensive cattle production systems. Differences in net profit between herds with tick control compared to those without were US$22,619, US$13,902 and US$28,290 for the extensive, semi-intensive and intensive production systems, respectively [190]. Utilizing these threshold calculations can guide profitable acaracide use.

## 4. Implementing Parasite Control Measures

Clearly, the parasite infection of cattle results in lost productivity with enormous financial losses to cattle producers. Parasiticides can be employed therapeutically, to treat parasitism resulting in clinical disease, for production gains, to improve reproductive performance, and to prevent future infection [65]. The emergence of anthelmintic and acaricidal resistance begs the question, can parasiticides be used effectively and economically? With the changing sensitivity of parasites to anthelmintics, endectocides and acaricides, cost effective treatments must be devised that provide positive return on investment considering all input costs. A threshold of parasitism, above which treatment outcome results in positive return on investment but below which non-treatment is the more prudent response, must be determined. For example, with mild infection, cattle may not show any measurable adverse effects. Above a certain threshold of infection, economic impact occurs due to reduced production results. This reduction in production may not be obvious and may be difficult to measure [46]. Yet to be profitable, producers will need to determine when and how to implement parasite control measures.

## 5. Parasite Control by Production Type

Parasitic exposure, infection and disease will differ by geography, class of cattle and associated management practices. The goals for treating parasites in beef cattle differ depending upon the goals of the producer. In some regions of the world the goal is simply to prevent mortality. In other regions of the world, where producers have already implemented controls necessary to maximize survival, the goal is to implement parasite control practices that optimize productivity. Resources for mitigating the effects of parasitism will also differ by region and type of production. Even within a given region, no single parasite control strategy will be suited for all herds [46].

Selection and use of anthelmintics and acaricides based on parasite prevalence and resistance profile, along with the appropriate timing of administration, are key to successful implementation. Fecal egg count monitoring and nematode identification via PCR can be valuable for guiding chemotherapy selection [144]. For tick control, suspected acaricide resistance should be confirmed in the laboratory, when possible, with assays such as larval packet test (LPT), for example [79,82,83]. Cattle producers in each production sector, cow-calf, stocker/pasture raised and feedlot can adopt sector-appropriate parasite control strategies to enhance cattle health, performance and profitability [24].

### 5.1. Cow-Calf

Irrespective of geography, all grazing cattle, including cow-calf herds, are exposed to gastrointestinal nematodes (GIN) that contribute to productivity loss [43]. For cow-calf operations, productivity is measured by pregnancy rate, average daily gain (ADG) and calf weaning weight [24]. Treatment to control GIN in heifers and cows has been associated with productivity gains in cow-calf herds and replacement heifers [14,18,152,191,192]. In a review of 170 research studies to determine the economic benefit of various pharmaceutical technologies including parasite control, growth promoting implants, sub-therapeutic antibiotics, ionophores, and beta agonists, based on US pricing in 2005, Lawrence et al. (2007) found that 73% of cow-calf operators utilize de-wormers and 81% utilize fly control. Their analysis found that parasite control in the cowherd has a significant positive impact on calf production, with de-wormers increasing weaning rate by 23.6% [24]. For example, Stromberg et al. (1997) showed, over a two-year study, that cows treated with fenbendazole at spring turnout and re-treated along with their calves in midsummer had calves that significantly outgained the control calves in both years. The average daily gain (ADG) for calves and reproductive performance for cows were both significantly greater than for control cattle. The pregnancy rate averaged across both years was 94% for the treated cows compared to 82% for the control animals [13]. While return on investment was not calculated in this study, the advantage of the ADG, the value of kilograms of calf-weight sold, the number of days on pasture, anthelmintic the cost of the product and an appropriate cattle-handling charge can be used to calculate the return. The results also indicated significantly reduced parasite egg shedding and pasture contamination.

The strategic use of anthelmintics can also be used to influence reproductive performance in estrus-synchronized cows [17,152]. The economic efficiency of parasite control over long- versus short-term periods (PC-LT vs. PC-ST) on reproductive performance of estrus-synchronized Angus cross beef cows in North America was studied by Johnson et al. (2020). The authors found that long-term parasite control effectively reduced parasite load, maintained or gained body condition and contributed to improved pregnancy outcomes. [17]. Similarly, in Brazil the use of anthelmintics with estrus synchronization and timed artificial insemination (TAI) was shown to increase pregnancy rate following the first TAI and elicit positive ROI in Nelore cows on pasture naturally infected with GIN [152].

Where exposure to liver fluke is high, adding flukicide to nematode control has been shown to produce higher condition scores and weight gains than when heifers were treated for nematodes alone [14]. In temperate regions, the recommended production-based treatment threshold for liver fluke is an individual FEC greater than five eggs per gram of feces (EPG) and for herds with greater than 25-percent prevalence [65].

### 5.2. Weanling-Stocker-Pasture Cattle

Stocker cattle and those finished on pasture are routinely exposed to GIN. Therefore, it is not surprising that of the available production-enhancing pharmaceuticals, including parasite control, growth-promoting implants, sub-therapeutic antibiotics, ionophores and beta agonist, de-wormers affect average daily gain (ADG) the most in stocker operations [24]. Numerous studies have shown that anthelmintics used in first- and second-season grazing cattle contribute to increased weight gain ranging from 11.85 to 49 kg per animal, when compared to non-dosed animals [99,149,193,194]. Knowledge of regional nematode prevalence and anthelmintic efficacy is critical for a successful outcome and positive ROI. For example, no advantages in average daily gain in grazing cattle treated once with 3.5% doramectin, 3.15% ivermectin or 1% doramectin compared to non-treated cattle resulted when nematodes were resistant to avermectins [99]. With cattle exposed to ML-resistant nematodes on pasture, Walker et al. (2013) demonstrated greater body weight and ADG in calves strategically dewormed with oxfendazole, with or without moxidectin, compared to untreated calves, and improvements for the combination versus treatment with moxidectin alone [143]. Studies such as these demonstrate the importance of garnering technical knowledge of parasites present, information beyond simple FEC, to gain positive returns on investment. Since nematode species such as *Haemonchus* spp. are more pathogenic than *Cooperia* spp., the identity of both parasite species present and the resistance profile is needed. Strategies such as targeted treatment or targeted selective treatment may help slow AR [140,141,194,195,196,197,198]. Increased costs associated with the testing necessary to implement TST must be considered [140]. To be effective, each farm will need to consider local practices and epidemiological and economic factors to devise the optimal treatment strategy for a specific farm [139].

The timing of anthelmintic use is also critical for optimizing GIN control and ROI. In central Brazil, the strategic control of GIN in beef cattle during the growing phase, from weaning up to 18–24 months of age when helminth susceptibility is high, can reduce parasite load and environmental contamination. Studies have shown that three anthelmintic treatments administered in May, August and November were more effective for weight gain than three treatments in May, July and September and more effective than two treatments administered in May and November [145,155]. Proper nutrition and anthelmintic selection can also impact ROI. Ramos et al. evaluated the effects of different anthelmintics on FEC, weight gain and overall economic efficiency in *B. taurus* x *B. indicus* beef cattle, 7–9 months of age, naturally infected with GIN, intensively reared on pasture with supplemental feed, on four farms in Rio Grande do Sul state in Brazil. Economic efficiency was calculated considering the cost of treatment, including associated labor and profit from additional cattle weight gained. Calves were treated in March 2017 with either ivermectin 1%, ivermectin 3.15%, eprinomectin 5%, levamisole 7.5%, albendazole 15% or no treatment and evaluated over 150 days. Levamisole 7.5% presented the best capacity for the reduction of nematode eggs in all herds, followed by albendazole 15% and eprinomectin 5%. On two farms, the best economic performance resulted from levamisole treatment; on another ivermectin 3.15% was best. Interestingly, on the fourth farm, no treatment was most profitable. Parasite resistance to multiple drugs, including MLs, was found in all herds, which likely contributed to the variable outcomes, along with variables not measured. This study that suggests proper nutrition along with an effective anthelmintic treatment, one that efficiently reduces EPG, will lead to production and economic returns [85].

Nematode control in grazed stocker cattle benefits cattle at the feedlot as well; cattle effectively treated for GIN prior to feedlot arrival delivered numerically greater total income per steer than did cattle arriving with high GIN egg counts despite deworming on arrival at the feedlot [67,150,199]. Steers dewormed at the feedlot did respond to anthelmintic intervention, but they did not experience compensatory gain during the feedlot phase and tended to have altered carcass composition and reduced marbling scores at slaughter [150]. Yazwinski et al. (2015) also found fewer illnesses during time in the feed yard for feedlot cattle that were treated for GIN while on the pasture prior to arrival at the feedlot [67]. These studies show that GIN control in growing cattle might be economical for both grazer and feedlot operators and provide further justification for their implementation.

Where ticks are prevalent, adding tick control to nematode control contributes to production gains and financial returns [7]. Bianchin et al. (2007) showed that the treatment of *B. taurus* x *B. indicus* steers in a subtropical region of Brazil Cerado with three anthelmintic treatments, alternating albendazole and doramectin or doramectin alone, during the winter, and three fipronil insecticidal-acaricidal treatments during the spring/summer provided significant additional weight gain. Steers treated for GIN gained a mean of 33 kg more than untreated steers. Steers additionally treated for ectoparasites had additional mean weight gains of 13 kg compared with non-treated steers [7].

In pastured and silage-supplemented, 8- to 9-month-old *B. taurus* x *B. indicus* cattle co-parasitized with *R. microplus, H. irritans* and GINs in subtropical southeastern Brazil, Gomes et al. (2022) found the treatment protocol utilizing an ectoparasiticide plus an endoparasiticide showed better outcomes with regard to parasite counts, productivity and financial data than strategic treatment using an endectocide alone [153]. In this study, over 308 days, cattle were either treated four times with fluazuron + fipronil and twice with fenbendazole, or cattle were treated four times with ivermectin + abamectin. At the end of the 308-day study, the group treated with fluazuron + fipronil and fenbendazole had significantly greater rate of weight gain (*p* < 0.05), total weight gain (*p* < 0.05) and return on investment compared to cattle treated with ivermectin + abamectin. While input costs (labor and product) for the group treated with the ecto- and endoparasiticides was 1.6 times higher than costs for the endectocide treated group, cattle in this group gained 15.4 kg more and provided a comparative return on investment (ROI) of 15.8. While strategic treatment with endectocides is popular in Brazilian cattle production for its convenience, this study demonstrates that financial returns from treatment with an ectoparasiticide along with an endoparasiticide can deliver better ROI.

### 5.3. Feedlot

Parasiticides, avermectins and fly control are the most commonly used pharmaceuticals in feedlots in the United States [24]. Cattle destined for the feedlot invariably come from pastures where they are exposed to GIN; therefore, cattle are oftentimes treated for these parasites upon feedlot arrival. Since nematode transmission is not a problem in the feed yard, short-acting anthelminitics such as oral fenbendazole have been shown to improve weight gain and carcass grade with significant financial returns [154]. MLs, combined with a benzimidazole or imidazothiazole, are also commonly used [67]. Adding a benzimidazole or imidazothiazole along with the ML removes nematodes resistant to the ML while the ML removes arrested nematodes, such as *O. ostertagi*, and controls lice, flies and ticks. With this treatment, production parameters (ADG, feed efficiency [FE], and dry matter intake [DMI]) have been shown to improve for cattle whether they were previously treated or not. However, improvements in ADG, FE, and DMI were greater in stocker cattle previously untreated, with increases of 13.4%, 4.9% and 7.1%, respectively, wherein stocker cattle that were previously treated while on pasture had improvements of 4.2%, 0.3% and 2.8%, respectively [67].

In the feedlot, cattle infected with liver fluke have decreased ADG and FE, poorer carcass quality scores, and bring lower prices gains compared to uninfected cattle [156,157,158,199]. However, by the time cattle previously exposed to liver fluke arrive at the feedlot, damage from liver fluke infection has already been inflicted; therefore, treatment with flukicide at this stage is oftentimes futile [67,200].

The economic insights into the impact of parasitism in beef cattle by parasite category and production type are summarized in Table 3 and Table 4, respectively.

## 6. Conclusions

Parasitism in beef cattle negatively impacts liveweight, feed efficiency, reproduction, calf yield and carcass quality, is a leading cause of liver condemnations and may be implicated in the increase production of greenhouse gases. These production losses can be significant and negatively impact the quantity of meat produced necessary to feed an increasing population. Losses increase the cost of meat production and result in diminished financial returns for beef producers while contributing to alterations in the atmospheric environment and climate. However, implementing strategic parasite control measures, with thorough knowledge of parasite risk, prevalence, parasiticide resistance profiles and prices, can result in positive economic returns for beef cattle farmers in all sectors and partially address concerns associated with the agricultural production of greenhouse gases.

## Figures and Tables

**Table 1 animals-13-01599-t001:** Common cattle parasites of economic importance.

Examples of Parasites of Cattle Common Name (*Scientific Name*)
INTERNAL PARASITES
Gastrointestinal Nematodes (GIN)
brown stomach worm (*Ostertagia ostertagi*)
small stomach worm (*Trichostrongylus axei*; *T. colubriformis*)
small intestinal roundworm (*Cooperia* species)
nodular worm (*Oesophagostomum radiatum*)
barbers pole worm, wire worm (*Haemonchus placei)*
long-neck worm, tread-necked worm (*Nematodirus helvetianus)*
cattle hookworm, nodular worm *(Bunostomum phlebotomum)*
Respiratory Nematodes
lungworm (*Dictyocaulus viviparus*)
Trematodes (Flukes)
liver flukes (*Fasciola hepatica*)
giant liver fluke (*Fasciola gigantica*)
rumen fluke (*Paramphistomatidae* spp.)
EXTERNAL PARASITES
Ticks
tropical cattle tick, or cattle fever tick (*Rhipicephalus* (*Boophilus*) *microplus*)
brown ear tick (*Rhipicephalus appendiculatus*)
tropical bont tick (*Amblyomma hebraeum*)
lone star tick (*Amblyomma americanum)*
cayenne tick *(Amblyomma cajannense)*
castor bean tick *(Ixodes ricinus)*

**Table 2 animals-13-01599-t002:** Economic burden of parasites. Estimated annual losses from parasites in beef cattle ^1,2^.

Country	GIN	Fluke	Tick	Internal Parasites and Ticks	References
Brazil	US$7.11 B	US$0.210 B	US$3.24 B	n/a	[20,21]
Mexico	US$0.45 B	US$0.13 B	US$0.57 B	n/a	[9]
Europe ^3^	€0.423 B (includes fluke)	n/a	n/a	n/a	[22]
Australia	AUS$0.0 936 B	n/a	AUS$0.161 B	AUS$0.2546 B	[23]
USA	US$8.5 B ^4^	US$0.00116 B ^5^	n/a	n/a	[19,24,25,26,27]

n/a = not available.^1^ Cattle parasites causing the greatest economic losses in beef cattle production worldwide are nematodes, trematodes and ticks. Table 1 identifies common cattle parasites of economic importance. ^2^ B = billions in currency. ^3^ Data derived from 18 European Union and near neighboring countries including Austria, Belgium, France, Germany, Ireland, Israel, Italy, Lithuania, Netherlands, North Macedonia, Norway, Poland, Portugal, Romania, Spain, Sweden, Tunisia, and the United Kingdom [22]. ^4^ Value given is the product of the estimated increased production cost per head due to GIN (US$190) and the number of US adult cattle marketed in 2020 (44,904,700) [19,24]. ^5^ Value is the product of the estimated cost of offal condemnation at slaughter (US$2.56 per head) and the number of livers condemned due to liver fluke in US slaughtered cattle in 2016 [25,26,27].

**Table 3 animals-13-01599-t003:** Economic insights into the impact of parasitism in beef cattle by parasite category.

Parasite Control in Beef Cattle	Parasite control practices can optimize productivity and financial returns. Within a given region, no single parasite control strategy will be suited for all herds [46]. Selection and use of anthelmintics and acaricides must be based on parasite prevalence, resistance profile and appropriate timing of administration.
Nematode Control	Nematode infection negatively impacts feed intake, growth rate, carcass weight, carcass composition, fertility, immune response and milk yield [43,117]. GIN Infection oftentimes causes subtle changes in production efficiency that are difficult to recognize, but which result in economic losses [118,119]. Anthelmintic treatment can improve reproductive performance with significant increases in pregnancy rate in anthelmintic treated cows and heifers [13,17,18,120]. Proper nutrition along with anthelmintic treatment that effectively reduces nematode egg shedding treatment can yield better cost effectiveness from improved animal production indices [86]. Due to worldwide anthelminthic and macrocyclic lactone resistance in nematodes of beef cattle, strategies that employ timely combinations of drug therapy have been reported with greater success [67,93,143,144,145,146]. Knowledge of the unique anthelmintic resistance landscape on individual farms will be necessary for devising economically sound deworming strategies [57,142].
Trematode Control	Prevalence in cattle is variable, with ranges of 1.2–91% in Africa, 3–67% in the Americas (ex-USA), 1–69% in Asia, 26–81% in Oceania and 0.12–86% in Europe [114]. Fasciolosis causes huge financial losses to butchers, farmers and consumers in the form of liver condemnation, poor quality carcasses, reduction in growth rate and reduced productivity [114]. Fasciola spp. often causes subclinical disease with resultant reductions in feed efficiency, growth, fertility (delayed puberty and increased calving interval) and overall loss of productivity [45,64,68,156,159,160]. In a study of 160 Scottish farms, liver fluke infection caused an average 6% reduction in profitability on an average beef farm [77]. Worldwide annual losses in cattle due to liver fluke infection have been estimated at US$3.0 billion [94,114].
Tick Control	About 80% of the world’s cattle are affected by ticks and tick-borne diseases (TBD), both of which cause significant production losses [112]. Ticks cause severe economic losses through the direct effect of tick attachment and resultant “tick worry” by vectoring of disease-causing pathogens such as *Babesia* [34,42,61,172]. Estimates place bovine babesiosis at the top of arthropod-borne diseases causing financial losses for cattle producers [11,34]. In Brazilian beef cattle, tick infestation resulted in estimated weight loss equivalent to US$34.61 per animal in the backgrounding phase and US$7.97 per animal in the finishing phase [172]. Routine and indiscriminate use, as well as inaccurate dosing, have contributed to acaricide resistance worldwide, including cross-resistance and multiple drug resistance (MDR), especially in *R. microplus* [82,95]. Rotational use of acaricides with differing mode of action and use of acaricide combinations along with tick population monitoring and pasture management may help preserve the efficacy of existing compounds [188,189]. A treatment protocol in Brazil utilizing an ectoparasiticide plus an endoparasiticide showed better outcomes with regard to parasite counts, productivity and financial data than strategic treatment using an endectocide alone [153]. Tick control, irrespective of treatment class (organophosphates, pyrethroids, or ML) or method of application (spray, pour-on or injection), was financially rewarding [172].

**Table 4 animals-13-01599-t004:** Economic insights on the impact of parasitism in beef cattle by production type.

Cow/Calf Production	Cost-saving on production was greatest for cow–calf operations that used dewormers, with savings of US$165 per head [24]. Parasite control in US cowherds has a significant positive impact on calf production, with de-wormers increasing weaning rate by 23.6% [24]. In a large survey of cattle producers in the USA, control of liver fluke resulted in 8 to 10 kg heavier cull cows, 1% to 3% more calves, and 14 to 20 kg heavier calves at weaning, yielding a net return to the producer of US$15.19 to US$31.03 per brood cow, depending on size of the calf crop and calf prices [165].
Stocker Production	Elimination of deworming resulted in increased cattle production cost of US$190 per head over the lifetime of the animal, based on 2005 US production costs and sale prices [24]. Numerous studies have shown that anthelmintics used in first and second season grazing cattle contribute to increased weight gain ranging from 11.85 to 49 kg per animal, when compared to non-dosed animals [99,149,193,195]. In Brazilian grazed cattle, Conde et al. (2019) found that each US$1 spent on deworming, including product and labor costs, resulted in a return of US$157 and US$134 using two or three annual doses of anthelmintic, respectively [137]. Treating heifers for both nematodes and liver fluke resulted in higher body condition scores and pregnancy rates than heifers treated for nematodes alone [14].
Feedlot Production	Cattle effectively treated for GIN prior to feedlot arrival delivered numerically greater total income per steer than did cattle arriving with high GIN egg counts despite deworming on arrival at the feedlot [150]. Since nematode transmission is not a problem in the feed yard, short-acting anthelmintics such as oral fenbendazole have been shown to improve weight gain and carcass grade with significant financial returns [154]. Eliminating de-wormer use in stocker and feedlot cattle resulted in increased production costs of US$21 and US$22 per head, respectively [24]. GIN treatment of Brazilian feedlot cattle with fenbendazole resulted in a net return of US$14.60 per animal [154].

## Data Availability

No new data were created or analyzed in this study. Data sharing is not applicable to this article.

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
