# Peer review of "The Economic Impact of Parasitism from Nematodes, Trematodes and Ticks on Beef Cattle Production"

_animals, 2023, doi:10.3390/ani13101599_

Round 1
Reviewer 1 Report
The large population growth requires consumption of more meat such as beef to meet protein intake. There is no doubt that parasites are a serious threat to the development of the beef cattle industry. Studies have shown that parasites not only reduce the performance of beef cattle, but also negatively affect immunity, metabolism, and many other aspects. In addition, some zoonotic parasitic diseases may also threaten human health. Therefore, research on parasites is crucial to the development of the beef cattle industry. This manuscript is very interesting. However, some simple revisions are required before publication.
Please add some background in the abstract.
The background, significance and purposes of this study need to be discussed in more detail.
Many conclusions in the manuscript lack relevant references. Please add.
Quality of English is ok.
Author Response
The economic impact of parasitism from nematodes, trematodes and ticks on beef cattle production. Tom Strydom, Robert P. Lavan, Siddhartha Torres and Kathleen Heaney.
For ANIMALS journal
May 1, 2023
Reviewer 1
The large population growth requires consumption of more meat such as beef to meet protein intake. There is no doubt that parasites are a serious threat to the development of the beef cattle industry. Studies have shown that parasites not only reduce the performance of beef cattle, but also negatively affect immunity, metabolism, and many other aspects. In addition, some zoonotic parasitic diseases may also threaten human health. Therefore, research on parasites is crucial to the development of the beef cattle industry. This manuscript is very interesting. However, some simple revisions are required before publication.
Please add some background in the abstract.
Author's Response: Added
The background, significance and purposes of this study need to be discussed in more detail.
Author's Response: Additional background, significance and purpose added to the “Introduction”.
Many conclusions in the manuscript lack relevant references. Please add. Author's Response: Additional references added.
Author Response
The economic impact of parasitism from nematodes, trematodes and ticks on beef cattle production. Tom Strydom, Robert P. Lavan, Siddhartha Torres and Kathleen Heaney.
For ANIMALS journal
May 1, 2023
Responses to Reviewer 2
General concept comments
These types of reviews are inherently periodically required and relevant as the situation is always changing over time. This paper, as such, fills the gap of providing the most up-to-date information in one place for ease of use for furthering other researchers' efforts. I can see this being highly applicable for many researchers as a citation towards securing grants and validating their research into parasitism in beef cattle and livestock overall. I do wonder, however, if there is an opportunity to include some information on the scope of parasite control on climate change directly. By this, I refer to the agreed understanding, in research, that healthier more productive animals have less carbon dioxide equivalent (CO2e) outputs per kg of product. As agriculture, and society as a whole, moves towards a more sustainable future where carbon credits and environmental subsidies in food production will be increasingly present. This will likely add to the economic impact considerations of parasitism. In such scenarios, not only would you be modelling the economic return of investment of parasite treatment with regards to productivity gains, but also mitigated CO2e emissions. As you have a lot of information cited, across various papers, regarding production gains relating to parasite treatment, it would be a simple case of finding some references to CO2e emissions per day in various beef systems. To provide some rough estimation of CO2 e emission reductions achieved via treatments which reduce time to slaughter. Equally, CO2e values per Kg product would be influenced by the papers you cited which improve slaughter carcass weights and figures for the impacts of this would likewise be easy to incorporate. It could be as simple as a table including the data from the paper production gains you cited vs their CO2e impacts. To make it more directly economic this could then be linked to estimated values for mitigating X kg/tons of CO2 e. A section like this would map well to the journal's focus in areas including sustainability of animal systems. Adding interest to beef producers achieving an extra possible revenue stream; particularly as in your own words, “cattle operations face narrow operating margins”. It would also likely improve the wider relevance and citation of the paper for future sustainable livestock production research and grants.
Author’s Response: Thank you for this inciteful suggestion. The impact of parasite control on livestock greenhouse gas emissions could result in further financial returns to producers. This is not something we had previously considered. There does seem to be evidence that controlling GIN and liver flukes results in reductions in GHG emissions [1-5]. However, the issue appears to be quite complex and therefore beyond the scope of this paper and the expertise of its authors [6]. Nonetheless, we have added mention of the potential impact of cattle parasite infection on greenhouse gas emission, with references, should our readers be inclined to research the topic.
The sentence starting line 28 seems counterintuitive. Why are you talking about eliminating parasite control? The sentence is very unclear in how it reads and why this notion is being brought in.
Author’s Response: Edited as recommended.
A lot of the information in lines 53 to 77, whilst perfectly accurate, is summed up in the table.2 so seems inefficient to be duplicating information – Maybe stick to just the contextual information surrounding the economic figures in the text.
Author’s Response: Edited to remove some of the detail. However, we find the number quite compelling. We suggest keeping the initial breakdown of losses per parasite for Brazil to “tee up” the enormity by parasite species, but then just list the total losses for the other countries with the detail breakdown in Table 2.
Good contextualisation of relevance in lines 118-119.
Author’s Response: Thank you.
Line 152-153 I would query if pasture burning for the reason of parasite control should be highlighted in a journal aiming for a focus on sustainability – It has more value for the prevention of wildfires surely.
Author’s Response: The mention of controlled pasture burning for parasite control was deleted. However, pasture burning as a means to limit parasite exposure is recommended (by Kumar et al 2013, as referenced in this manuscript [reference 58]. Pasture burning for parasite control is also mentioned in other publications not referenced here. As this is a review of published works, we thought mention of pasture burning should be included. Furthermore, in areas of cattle production where pastures are burned to enhance forage growth, parasite control is an added benefit. However, we have deferred to preferences of Reviewer 2 and deleted this fact.
242 – 245 I would again avoid repeating statistics just reference to table.2 and put the contextualising text for the figures in only, otherwise it feels repetitive.
Author’s Response: Edited to remove repeated statistics.
267-268 Highly valid point. Good to raise this in context – Nice!
Author’s Response: Thank you.
362-368 Again just reference Table.2 and focus on any new information in the text only. Author’s Response: Edited to remove some statistics.
524 – I would question whether this section is titled appropriately it also feels like something which should be part of a conclusion rather than in the main body of the text – particularly as it poses so many questions for future research areas. It also doesn’t particularly flow from section 3 or into section 5 so moving it wouldn’t impact the rest of the text.
Author’s Response: This section was reworded to make clearer its intent as a transition to the discussion of parasite control.
553-555 The reference is very broad and not parasite-specific maybe provide more insight with specific references or just remove this section. Its also an older text so the likelihood that many of these technologies have been superseded is fairly high.
Author’s Response: This sentence was meant as an introduction to the following section. It has been reworded accordingly.
559- 561 Tables 3 and 4 – Very well-curated summarisation tables but I don’t think it fits here, especially since it summarises information from sections that have yet to be discussed in the text, namely the production types. I would shift this to after the production type sections as a pre conclusion summarisation (or potentially even to the start of the text as an executive summary).
Author’s Response: Tables 3 and 4 were moved to follow the sections on parasite control by production sector.
Starting at line 562 Sections 6., 7. and 8. should really be sub-topics of section 5 and relabelled 5.1, 5.2 and 5.3 (or 4.1, 4.2, and 4.3 if you follow my advice to move section 4 to the conclusion) as they all fall under the main topic heading of ‘’Parasite Control by Production Type’.
Author’s Response: Subsections were created as suggested.
New References Added:
- Fox, N.J.; Smith, L.A.; Houdijk, J.G.M.; Athanasiadou, S.; Hutchings, M.R. Ubiquitous parasites drive a 33% increase in methane yield from livestock. International Journal for Parasitology 2018, 48, 1017-1021, doi:https://doi.org/10.1016/j.ijpara.2018.06.001.
- ADAS. Study to Model the Impact of Controlling Endemic Cattle Diseases and Conditions on National Cattle Productivity, Agricultural Performance and Greenhouse Gas Emissions Final Report. Available online: https://randd.defra.gov.uk/ProjectDetails?ProjectId=17791 (accessed on 18-January-2023).
- Kenyon, F.; Dick, J.; Smith, R.I.; Coulter, D.G.; McBean, D.; Skuce, P.J. Reduction in Greenhouse Gas Emissions Associated with Worm Control in Lambs. Agriculture 2013, 3, 1-14.
- Jonsson, N.N.; MacLeod, M.; Hayward, A.; McNeilly, T.; Ferguson, K.D.; Skuce, P.J. Liver fluke in beef cattle – Impact on production efficiency and associated greenhouse gas emissions estimated using causal inference methods. Preventive Veterinary Medicine 2022, 200, 105579, doi:https://doi.org/10.1016/j.prevetmed.2022.105579.
- Hristov, A.N.; Ott, T.; Tricarico, J.; Rotz, A.; Waghorn, G.; Adesogan, A.; Dijkstra, J.; Montes, F.; Oh, J.; Kebreab, E.; et al. Special topics--Mitigation of methane and nitrous oxide emissions from animal operations: III. A review of animal management mitigation options. J Anim Sci 2013, 91, 5095-5113, doi:10.2527/jas.2013-6585.
- Charlier, J.; Morgan, E.R.; Kyriazakis, I. Quantifying the Interrelationship between Livestock Infections and Climate Change: Response to Ezenwa et al. Trends Ecol Evol 2021, 36, 576-577, doi:10.1016/j.tree.2021.02.003.
Reviewer 3 Report
In my opinion this is a very interesting, very well structured and updated review about the economic impact of three of the most representative groups of parasites affecting beef cattle production. This review comments about important information about direct or indirect loses occasioned by these groups of parasites and the use of anthelmintic drugs, anti-parasitic resistance in nematodes, trematodes and ticks in different production systems, cost benefits of using anti parasitic treatments in health and production of beed cattle herds.
I suggest to insert (somewhere) a brief paragraph commenting about the use of combination of drugs either anthelmintic or ixodicides in relation to the potential risk of multi-drug resistance; as well as the use of ML as endectoxides and its possible risk of anti-parasitic resistance gene spreading either in endo and ectoparasites.
I only found minimal details that should be fixed.
For instance, In Table 1 in the section of External parasites (Ticks) it said Amblyomma Americanum, Americanum should start with lower case;
in page 5, line 177 after ...host resistance, there is a dot. please delete it.
In page 6, line 243 if am ok, because economic losses is in plural, after fluke, should say "were", instead of "was" but I maybe wrong, so, please check it!
In my opinion this is a very god review that deserves to be published.
My recommendation is: Accepted after minor revision
Author Response
The economic impact of parasitism from nematodes, trematodes and ticks on beef cattle production. Tom Strydom, Robert P. Lavan, Siddhartha Torres and Kathleen Heaney.
For ANIMALS
May 1, 2023
Response to Reviewer 3
In my opinion this is a very interesting, very well structured and updated review about the economic impact of three of the most representative groups of parasites affecting beef cattle production. This review comments about important information about direct or indirect loses occasioned by these groups of parasites and the use of anthelmintic drugs, anti-parasitic resistance in nematodes, trematodes and ticks in different production systems, cost benefits of using anti parasitic treatments in health and production of beed cattle herds.
I suggest to insert (somewhere) a brief paragraph commenting about the use of combination of drugs either anthelmintic or ixodicides in relation to the potential risk of multi-drug resistance; as well as the use of ML as endectoxides and its possible risk of anti-parasitic resistance gene spreading either in endo and ectoparasites.
I only found minimal details that should be fixed.
For instance, In Table 1 in the section of External parasites (Ticks) it said Amblyomma Americanum, Americanum should start with lower case;
Author’s Response: This change was made.
in page 5, line 177 after ...host resistance, there is a dot. please delete it.
Author’s Response: False period removed.
In page 6, line 243 if am ok, because economic losses is in plural, after fluke, should say "were", instead of "was" but I maybe wrong, so, please check it!
Author’s Response: We made the change to make the subject and verb agree in number.
In my opinion this is a very god review that deserves to be published.
My recommendation is: Accepted after minor revision
Submission Date
31 March 2023
Date of this review
10 Apr 2023 23:35:47
Round 2
Reviewer 1 Report
It could be accepted in the current version.
Quality of English language is good.